# Impact of tear metrics on the reliability of perimetry in patients with dry eye

**Hideto Sagara**[1,2]☯*, **Tetsuju Sekiryu**[2]☯, **Kimihiro Imaizumi**[2], **Hiroaki Shintake**[2], **Urara Sugiyama**[2], **Hiroki Maehara**[2]

**1** The Marui Eye Clinic, Minamisoma City, Fukushima, Japan, **2** Department of Ophthalmology, Fukushima Medical University School of Medicine, Fukushima City, Fukushima, Japan

☯ These authors contributed equally to this work.
* hide1234@ruby.ocn.ne.jp

## Abstract

### Background

The application of artificial tears before performing perimetry can improve the reliability and results of perimetry in patients with glaucoma and dry eye (DE). However, the effects of ocular surface and tear film conditions on perimetry measurements and reliability have not been fully characterized.

### Methods

This prospective, cross-sectional, multicenter study investigated tear metrics in perimetry and assessed the relationships that existed among ocular surface condition, tear condition, and perimetry reliability. Forty-three eyes (43 patients) with DE disease according to the 2016 Japanese diagnostic criteria of DE and 43 eyes (43 subjects) of age- and visual field mean deviation-matched normal control subjects were studied. Perimetry was performed using the Humphrey Field Analyzer (30–2 SITA-Standard). Schirmer's test, strip meniscometry value, blink rate, tear film break-up time (TFBUT), fluorescein staining of ocular surface, and Dry Eye-related Quality of Life Score (DEQS) were measured. Blink rate was re-measured during perimetry. TFBUT and fluorescein staining were re-evaluated after perimetry. Perimetry reliability was evaluated with fixation loss, false-positive, and false-negative rates.

### Results

Blink rate during perimetry was significantly lower for both patients with DE and normal controls (both P<0.001). TFBUT after perimetry was significantly higher than before perimetry in patients with DE (P<0.001). Fluorescein staining of ocular surface was significantly increased in patients with DE and normal control subjects (P = 0.002 and P<0.001, respectively). Spearman correlation analysis revealed that blink rate during perimetry was negatively correlated with fixation-loss rate (r = -0.393, P = 0.009) in patients with DE.

**Data Availability Statement:** All relevant data are available in the Figshare Repository at https://doi.org/10.6084/m9.figshare.9771926.

**Funding:** The authors received no specific funding for this work.

**Competing interests:** The authors have declared that no competing interests exist.

## Conclusions

Performing perimetry was associated with a significant change in tear condition and ocular surface condition in both patients with DE and normal control subjects. The changes in tear condition and ocular surface condition may impact the reliability of perimetry in patients with DE.

## Introduction

Perimetry, particularly automated perimetry, is widely used to determine visual field loss and to confirm worsening of glaucomatous damage. Ocular surface disease occurs in approximately 15% of the general elderly population and is reported in 48–59% of patients with medically treated glaucoma [1]. The application of artificial tears before performing perimetry can improve the reliability and results of perimetry in patients with glaucoma and dry eye (DE) [2–4]. An association between ocular surface disorders and increased frequency of eye movement during perimetry in pre-perimetric glaucoma has been reported [5]. Therefore, it is necessary to examine the tear film and ocular surface conditions of patients who have, or are suspected of having, glaucoma before performing perimetry in order to acquire high perimetry reliability. Nevertheless, the effects of ocular surface and tear film conditions on perimetry measurements and reliability have not been fully characterized.

During perimetry, patients are instructed to closely watch visual targets. Thus, most patients try to concentrate during perimetry to avoid missing the targets. As a result of this concentrated viewing (i.e., "staring"), tear condition may change drastically during perimetry. These changes may affect the results and reliability of perimetry measurements. However, the effects of perimetry-induced changes on tear condition, and the effects of tear condition change on perimetry measurements have not been clarified. The Humphrey Field Analyzer (Carl Zeiss Meditec, Dublin, CA, USA) is used worldwide to monitor visual field damage in patients with glaucoma [6–10]. The Humphrey Field Analyzer allows measurement of parameters that indicate the reliability of perimetry, such as fixation loss (FL), false-positive (FP), and false-negative (FN) rates [11, 12]. The aim of this study was to assess the relationships between indices of perimetry reliability, ocular surface condition change, and tear metrics. These indices were compared in two groups: patients with DE and normal control subjects.

## Materials and methods

### Participants and ethics

Forty-three eyes of 43 patients with DE, according to the 2016 Japanese diagnostic criteria of DE [13] (23 males, 20 females; mean age ± standard deviation [SD], 71.7 ± 8.1 years), with a median (interquartile range [IQR]) Humphrey visual field mean deviation (MD) of -4.0 (-10.0 to -0.8 dB), and 43 eyes of 43 age- and MD-matched normal control subjects (28 males, 15 females; mean age ± SD, 68.9 ± 9.9 years) with a median (IQR) MD of -4.0 (-9.0 to -0.7 dB) were recruited for this prospective multicenter study. MD rate [14] reportedly affects the reliability of perimetry indices; therefore, the MD rates were matched in this study. All patients attending the glaucoma specialty outpatient clinic of Fukushima Medical University or the Marui Eye Clinic from May to November 2014 were considered. The study was approved by the Institutional Review Board/Ethics Committee of Fukushima Medical University and

conducted according to the tenets of the Declaration of Helsinki. All patients recruited for the study provided written informed consent.

Patients with DE and normal control subjects who had a follow-up of more than 6 months with at least three perimetry measurements, a best-corrected visual acuity of 20/25 or better, an intraocular pressure of ≤21 mmHg, and who were diagnosed or suspected of having glaucoma were included. In cases where perimetry was performed on both eyes, it was performed first in the patient's right eye. Only eyes in which perimetry was initially performed were examined. Both patients with DE and normal control subjects were excluded if they had a clinical history of lacrimal duct atresia, atopy, allergies, Stevens-Johnson syndrome, injuries (chemical, thermal, or radiation), contact lens use, or any other ocular or systemic disorders that could create a tear condition and/or ocular surface problems.

A total of 27 eyes (62.8%) and 29 eyes (67.4%) in patients with DE, and normal control subjects, respectively, were treated with anti-glaucoma eye drops. Additionally, 15 eyes (34.9%) in patients with DE and 14 eyes (32.6%) in normal control subjects were treated with lubricant eye drops on a daily basis for more than six months. The average test times of patients with DE and normal control subjects were 495.4 ± 107.8 s and 502.5 ± 98.8 s, respectively, with no significant difference between groups (P = 0.749).

## Examinations

Temperature and humidity of the examination room were adjusted to 25–27 ˚C and 50–60%, respectively, during perimetry. Perimetry was performed using the Humphrey Visual Field Analyzer II (model 750) with the Swedish interactive threshold algorithm (SITA) standard strategy, the 30–2 program, and a size III white stimulus on a white background (equivalent to 31.5 apostilbs). The FL, FP, and FN rates were used to assess the reliability of perimetry.

In 2016, the Asia Dry Eye Society and Dry Eye Society of Japan implemented new diagnostic criteria for DE disease that enabled diagnosis with only two positive items: subjective symptoms and decreased tear film break-up time (TFBUT) (≤5 seconds) [13]. In this study, DE was diagnosed according to the new diagnostic criteria. Dry Eye-related Quality of Life Score (DEQS) questionnaires were used to evaluate the symptoms associated with DE [15]. DEQS scores > 15 were defined as being symptom-positive for DE [16]. The DEQS is a self-evaluation method, which provides assessment of the effects of DE symptoms on quality of life in general, including a patient's mental health. The DEQS includes 15 questions and two sub-categories: impact on daily life and bothersome ocular symptoms. Each questionnaire was scored on a 4-point Likert scale ranging from 1 to 4, with a larger number indicating a greater burden. The final score was calculated using the DEQS formula by multiplying the sum of the score by 25 and then dividing the total by the number of questions answered. The summary score for the DEQS ranges from 0 to 100; the higher the score, the greater the disability. This questionnaire has been validated and is often used in clinical trials [15].

Tear metrics and ocular surface condition were evaluated using five parameters: 1) Schirmer's test value, 2) tear meniscus volume using a Strip Meniscometry (SM) Tube (Eco Electricity Co, Ltd., Fukushima, Japan) [17, 18], 3) blink rate, 4) TFBUT, and 5) fluorescein staining (graded by the van Bijsterveld [VB] score [0–9]) [19, 20]. The use of all eye drops, including anti-glaucoma eye drops, was stopped 4 hours before all examinations to avoid eye drop effects. Schirmer's test without topical anesthesia and lacrimal irrigation were performed on other days, within 20 days of perimetry. SM value, blink rate in 1 min, TFBUT, and VB scores were evaluated 30 mins before starting perimetry. TFBUT and VB scores were re-evaluated immediately after the first perimetry. Expert optometrists observed the eyes of subjects and manually counted the number of blinks before perimetry, while subjects were in a relaxed

atmosphere. The participants were unaware that their blink rate was being observed. For measurement of TFBUT, the interval between the last complete blink and the first appearance of a dry spot or disruption in the tear film was recorded after the instillation of fluorescein. TFBUT was evaluated three times, and the mean value was recorded. The number of blinks during perimetry were recorded using the video eye monitor of the Humphrey Visual Field Analyzer II [21]. One complete closure of the eyelid constituted one blink [22]. The patients and subjects were divided into four groups, and we assessed the relationship between indices of reliability of perimetry, ocular surface conditions, and tear metrics. Group 1 comprised 35 eyes in 35 patients with DE with blink rates that decreased during perimetry. Group 2 comprised 8 eyes in 8 patients with DE with blink rates that increased or remained unchanged during perimetry. Group 3 comprised 35 eyes in 35 normal control subjects with blink rates that decreased during perimetry. Group 4 comprised 8 eyes in 8 normal control subjects with blink rates that increased or remained unchanged during perimetry.

### Statistical analysis

The differences in the blink rate before and during perimetry, and the differences in TFBUT and VB scores before and after perimetry, were assessed using the Wilcoxon signed-rank test. Differences in patient age were assessed using an unpaired *t*-test. Kruskal-Wallis *H* and Bonferroni correction tests were used to assess the differences in parameters among the four groups. Other values were compared using the Mann-Whitney *U* test. Correlations between the factors studied were assessed by Spearman nonparametric correlation analysis. A *P*-value $< 0.05$ was considered statistically significant. The data are presented as mean ± SD or median (IQR [min-max]). All statistical analyses were performed using R version 3.3.2 software (R Foundation for Statistical Computing, Vienna, Austria).

### Results

Schirmer's test values of $<10$ mm were observed in 38 eyes (88.4%) and 12 eyes (27.9%) in patients with DE and normal control subjects, respectively. In the DE group, only 1 eye (2.3%) had a normal SM rate ($>5$ mm), and in the control group, only one 1 eye (2.3%) also had a normal SM rate. Median blink rate during perimetry was significantly lower compared to pre-perimetry rates in both patients with DE: 5.2 (2.6–12.2 [0–39]) vs. 15.0 (9.5–19.0 [2–44]) /min and normal control subjects: 3.4 (1.4–8.2 [0–36]) vs. 14.0 (10.0–20.0 [3–48]) /min (both P<0.001; Table 1). Blink rates of 35 eyes (81.4%) decreased, and the rates of 8 eyes (18.6%) increased or remained unchanged during perimetry in both patients with DE and normal control subjects. There were no significant differences in perimetry reliability parameters between patients with DE and normal control subjects (Table 1). However, FL rate was significantly higher in eyes with decreased blink rates among patients with DE (P = 0.002; Table 2).

Pre-perimetry blink rates in the eyes with blink rates that decreased during perimetry varied from 3 to 44/min in patients with DE (Group 1) and from 4 to 48/min in normal control subjects (Group 3). Blink rates before perimetry in the eyes with blink rates that increased or remained unchanged during perimetry were $< 17$/min in patients with DE (Group 2) and 19/min in normal control subjects (Group 4), respectively (Fig 1). Blink rates during perimetry were significantly lower in Group 1 compared to Group 2, and in Group 3 compared to Group 4: Group 1, 4.6 (1.7–7.0 [0–21]) vs. Group 2, 13.4 (11.4–16.0 [2–39]) /min and Group 3, 2.6 (0.9–5.4 [0–12]) vs. Group 4, 20.1 (11.0–30.5 [4–36]) /min (P = 0.007 and P<0.001, respectively, Fig 1).

TFBUT after perimetry was significantly higher compared to that before perimetry in patients with DE: 4.0 (3.0–4.5 [1–5]) vs. 4.0 (3.0–5.0 [2–15]) sec (P<0.001; Fig 2). However,

**Table 1. Comparison of parameters in patients with dry eye and normal control subjects.**

| Tear condition and ocular surface parameters | Patients with dry eye | Normal control subjects | P-value |
|---|---|---|---|
| Schirmer's test (mm) | 4.0 (2.0–7.0[0.0–25.0]) | 14.0 (8.5–19.0[6.0–34.0]) | < .001* |
| SM value (mm) | 1.9 (1.5–2.7[1.0–5.5]) | 1.8 (1.5–2.7[1.0–7.0]) | .741 |
| Blink rate before perimetry (/min) | 15.0 (9.5–19.0[2–44]) | 14.0 (10.0–20.0[3–48]) | .969 |
| Blink rate during perimetry (/min) | 5.2 (2.6–12.2[0–39]) | 3.4 (1.4–8.2[0–36]) | .265 |
| TFBUT before perimetry (sec) | 4.0 (3.0–4.5[1–5]) | 8.0 (7.0–11.0[6–17]) | < .001* |
| TFBUT after perimetry (sec) | 4.0 (3.0–5.0[2–15]) | 8.0 (6.0–11.0[3–19]) | < .001* |
| VB score before perimetry | 1.0 (0.0–2.0[0–5]) | 0.0 (0.0–1.0[0–2]) | .046* |
| VB score after perimetry | 1.0 (0.0–3.0[0–6]) | 1.0 (0.0–2.0[0–5]) | .081 |
| DEQS | 35.0 (23.2–46.7[16.7–75.0]) | 8.3 (1.7–22.5[0.0–55.0]) | < .001* |
| **Perimetry Reliability Parameters** | | | |
| Fixation loss rate (%) | 11.1 (5.0–45.6[0.0–100]) | 11.0 (2.8–42.3[0.0–100]) | .931 |
| False positive rate (%) | 1.0 (1.0–4.5[0.0–62.0]) | 2.0 (0.0–8.0[0.0–81.0]) | .363 |
| False negative rate (%) | 6.0 (0.0–11.5[0.0–40.0]) | 7.0 (0.0–12.0[0.0–26.0]) | .705 |

Values are presented as median (interquartile range [min-max]). The Mann-Whitney *U* test was used to assess P-values.

*P<0.05. SM, strip meniscometry

TFBUT, tear film break-up time; VB score, van Bijsterveld score; DEQS, Dry Eye-related Quality of Life score.

there were no significant differences between TFBUT before perimetry and TFBUT after perimetry in normal control subjects: 8.0 (7.0–11.0 [6–17]) vs. 8.0 (6.0–11.0 [3–19]) sec (P = 0.562). In patients with DE, TFBUT was prolonged in some eyes with blink rates that did not decrease during perimetry, as well as in eyes with decreased blink rates during perimetry as follows: Group 1, 16 eyes (45.7%); Group 2, 6 eyes (75.0%); Group 3, 3 eyes (8.6%); and Group 4, 0 eyes (0%).

VB score significantly increased in patients with DE from 1.0 (0.0–2.0 [0–5]) to 1.0 (0.0–3.0 [0–6]), and in normal control subjects from 0.0 (0.0–1.0 [0–2]) to 1.0 (0.0–2.0 [0–5]) (P = 0.002 and P<0.001, respectively, Fig 3). There were no significant differences in VB score after perimetry between the eyes with blink rates that decreased during perimetry and the eyes with blink rates that did not decrease in patients with DE: Group 1, 1.0 (0.0–3.0 [0–5]) vs. Group 2, 2.0 (1.0–3.0 [0–6]), and normal control subjects: Group 3, 1.0 (0.0–2.0 [0–5]) vs. Group 4, 1.5 (0.8–2.0 [0–3]) (P = 0.407 and P = 0.378, respectively; Fig 3). In both patients with DE and normal control subjects, VB score was worsened in some eyes with blink rates that did not decrease during perimetry, as well as in eyes with decreased blink rates during perimetry as follows: Group 1, 9 eyes (25.7%); Group 2, 3 eyes (37.5%); Group 3, 11 eyes (31.4%); and Group 4, 3 eyes (37.5%).

SM value was directly correlated with FN rate (r = 0.341, P = 0.025), and blink rate during perimetry was negatively correlated with FL rate (r = -0.393, P = 0.009). DEQS was negatively correlated with FN rate (r = -0.323, P = 0.035) in patients with DE. However, there were no apparent correlations among the perimetry reliability parameters and other parameters in normal control subjects (Table 3).

## Discussion

Several studies have indicated that the use of anti-glaucoma eye drops increased the risk of DE [23, 24], and that the rate of DE in elderly patients was higher [25, 26]. Therefore, patients with glaucoma, especially elderly patients treated for a long period with anti-glaucoma eye drops,

**Table 2. Comparison of blink rate and other parameters during perimetry in patients with dry eye and normal control subjects.**

| | Blink rate during perimetry | | P-value |
|---|---|---|---|
| | Decreased | Increased or unchanged | |
| **Patients with Dry Eye** | Group 1 (n = 35) | Group 2 (n = 8) | |
| **Tear condition and ocular surface parameters** | | | |
| Schirmer's test (mm) | 3.0 (2.0–6.5[1.0–25.0]) | 5.0 (1.8–7.0[0.0–15.0]) | 1.000 |
| SM value (mm) | 1.8 (1.5–2.2[1.0–5.0]) | 2.8 (1.7–3.6[1.5–5.5]) | .102 |
| Blink rate before perimetry (/min) | 17.0 (10.5–19.0[3–44]) | 11.0 (8.8–13.3[2–16]) | .051 |
| Blink rate during perimetry (/min) | 4.6 (1.7–7.0[0–21]) | 13.4 (11.4–16.0[2–39]) | .007** |
| TFBUT before perimetry (sec) | 4.0 (3.0–4.5[2–5]) | 2.5 (2.0–4.3[1–5]) | .218 |
| TFBUT after perimetry (sec) | 4.0 (3.0–5.0[2–15]) | 3.5 (2.8–6.3[2–9]) | .621 |
| VB score before perimetry | 1.0 (0.0–2.0[0–5]) | 1.5 (1.0–2.3[0–3]) | .198 |
| VB score after perimetry | 1.0 (0.0–3.0[0–5]) | 2.0 (1.0–3.0[0–6]) | .407 |
| DEQS | 35.0 (22.5–47.5[18.3–75.0]) | 30.0 (24.6–36.3[16.7–46.7]) | .417 |
| **Perimetry reliability parameters** | | | |
| Fixation loss rate (%) | 13.6 (6.3–57.5[0.0–100]) | 2.2 (0.0–5.9[0.0–6.7]) | .002** |
| False positive rate (%) | 1.0 (1.0–5.5[0.0–62.0]) | 1.0 (1.8–2.5[0.0–45.0]) | .587 |
| False negative rate (%) | 6.0 (0.5–11.0[0.0–40.0]) | 4.5 (0.0–13.0[0.0–29.0]) | .788 |
| **Normal control subjects** | Group 3 (n = 35) | Group 4 (n = 8) | |
| **Tear condition and ocular surface parameters** | | | |
| Schirmer's test (mm) | 12.0 (8.0–21.0[6.0–34.0]) | 16.0 (10.0–19.0[7.0–21.0]) | .650 |
| SM value (mm) | 1.8 (1.4–2.6[1.0–4.5]) | 1.9 (1.7–3.6[1.5–7.0]) | .204 |
| Blink rate before perimetry (/min) | 16.0 (10.0–22.0[4–48]) | 10.0 (7.5–11.3[3–18]) | .036 |
| Blink rate during perimetry (/min) | 2.6 (0.9–5.4[0–12]) | 20.1 (11.0–30.5[4–36]) | < .001** |
| TFBUT before perimetry (sec) | 9.0 (7.0–11.5[6–17]) | 6.5 (6.0–9.0[6–11]) | .087 |
| TFBUT after perimetry (sec) | 8.0 (7.0–12.0[3–19]) | 6.5 (4.8–7.3[4–11]) | .044 |
| VB score before perimetry | 0.0 (0.0–1.0[0–2]) | 1.0 (0.0–1.3[0–2]) | .308 |
| VB score after perimetry | 1.0 (0.0–2.0[0–5]) | 1.5 (0.8–2.0[0–3]) | .378 |
| DEQS | 6.7 (1.7–18.4[0.0–55.0]) | 10.8 (3.8–25.8[0.0–41.7]) | .593 |
| **Perimetry reliability parameters** | | | |
| Fixation loss rate (%) | 18.2 (2.8–50.2[0.0–100]) | 7.9 (4.2–10.1[0.0–93.8]) | .386 |
| False positive rate (%) | 2.0 (0.0–8.0[0.0–81.0]) | 2.5 (2.0–5.3[1.0–11.0]) | .625 |
| False negative rate (%) | 8.0 (0.0–12.0[0.0–26.0]) | 1.0 (0.0–12.0[0.0–20.0]) | .526 |

Values are presented as median (interquartile range [min-max]). The Kruskal–Wallis *H* test and Bonferroni correction were used to access P-values.

**P<0.0083.

should be examined for DE. Tsubota et al. [27] reported that fully wet eyes do not depend on blinking to maintain a wet ocular surface, whereas desiccated eyes depend on blinking to maintain a wet ocular surface. The patients with DE whose blinking rates decreased during perimetry may have not been able to maintain a wet ocular surface. Subsequently, their visibility may have deteriorated during perimetry, and their ocular surface parameters may have worsened. As a result, it became difficult to observe the central fixation light of perimeter clearly [21], and in patients with DE, FL rates may have been higher in the eyes with blink rates that decreased during perimetry. However, in both patients with DE and normal control subjects, VB score was also worsened in some eyes with blink rates that did not decrease during perimetry, as well as in eyes with decreased blink rates during perimetry (Fig 3).

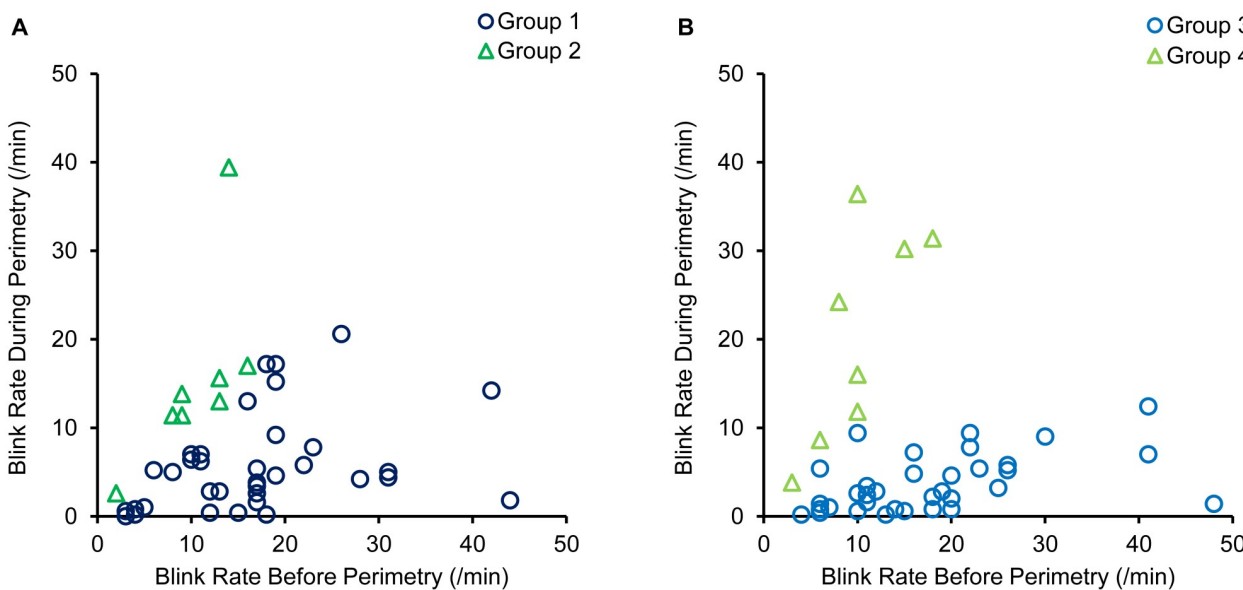

**Fig 1. Scatter diagrams of the relationships between blink rates before and during perimetry.** (A) Data for patients with dry eye. (B) Data for normal control subjects. Group 1; eyes with decreased blink rates during perimetry in patients with dry eye; Group 2; eyes with increased or unchanged blink rates during perimetry in patients with dry eye; Group 3; eyes with decreased blink rates during perimetry in normal control subjects; Group 4; eyes with increased or unchanged blink rates during perimetry in normal control subjects.

A wide range of blink frequencies occur during routine perimetry. Although some patients hardly ever blink during perimetry, others show frequent blinking that often appears to be related to stimulus presentation times [28]. The blinking rate is lower when mental load is higher. Conversely, the blinking rate is higher when mental load is lower [22]. The patients whose blinking rates did not decrease during perimetry may have been able to perform perimetry while relaxed, whereas patients whose blinking rates decreased during perimetry may

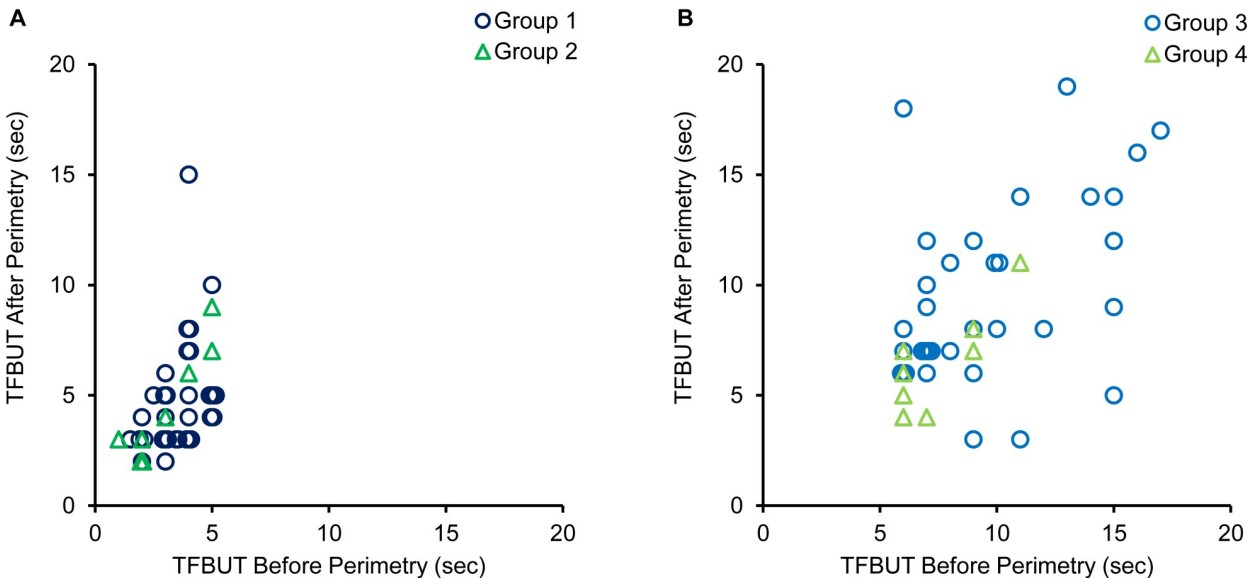

**Fig 2. Scatter diagrams of the relationships between TFBUT before perimetry and TFBUT after perimetry.** (A) Data for patients with dry eye. (B) Data for normal control subjects. TFBUT, tear film break-up time.

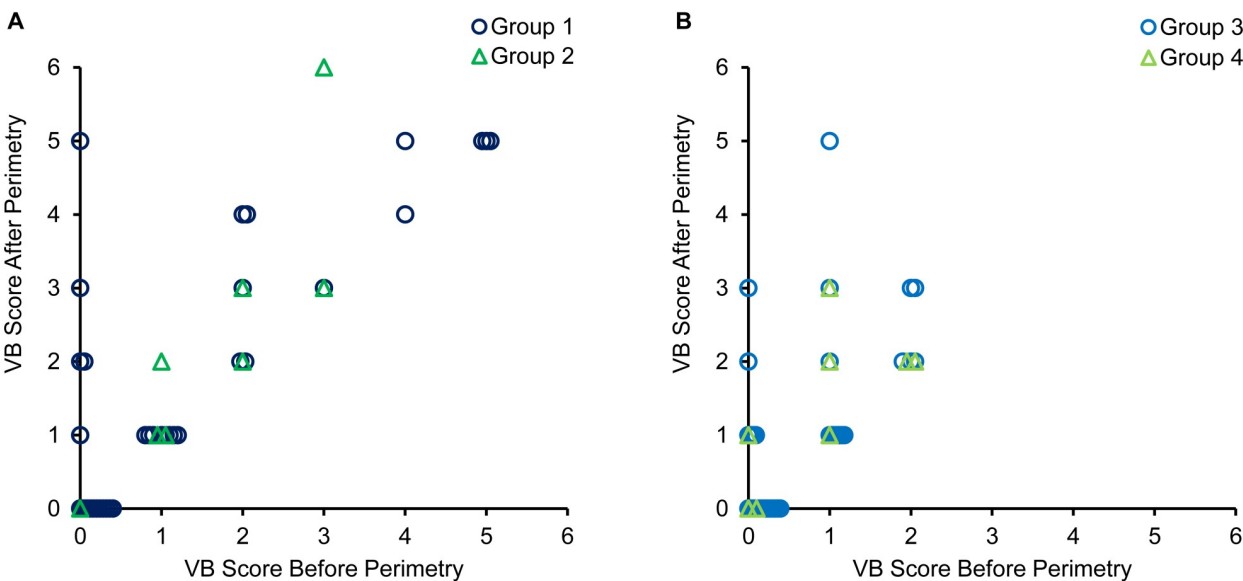

**Fig 3. Scatter diagrams of the relationships between VB score before perimetry and VB score after perimetry.** (A) Data for patients with dry eye. (B) Data for normal control subjects. VB score, van Bijsterveld score.

**Table 3. Spearman correlation coefficients between tear conditions, ocular surface findings, and perimetry reliability parameters in patients with dry eye and normal control subjects.**

| Tear condition and ocular surface parameters | Perimetry reliability parameters | | |
|---|---|---|---|
| **Patients with dry eye** | **Fixation loss rate** | **False positive rate** | **False negative rate** |
| Schirmer's test | -.263 | .125 | .159 |
| SM value | -.221 | .015 | **.341** |
| Blink rate before perimetry | .028 | .029 | -.101 |
| Blink rate during perimetry | **-.393** | -.098 | -.042 |
| TFBUT before perimetry | .258 | .120 | .265 |
| TFBUT after perimetry | .213 | .030 | .189 |
| VB score before perimetry | -.119 | -.240 | .246 |
| VB score after perimetry | -.247 | -.300 | .129 |
| DEQS | .144 | -.076 | **-.323** |
| **Normal control subjects** | | | |
| Schirmer's test | -.111 | .154 | .226 |
| SM value | -.066 | .058 | -.087 |
| Blink rate before perimetry | -.157 | -.190 | -.272 |
| Blink rate during perimetry | -.094 | -.126 | .119 |
| TFBUT before perimetry | -.012 | .103 | -.201 |
| TFBUT after perimetry | -.123 | -.188 | -.104 |
| VB score before perimetry | -.235 | -.062 | -.236 |
| VB score after perimetry | -.226 | -.090 | -.127 |
| DEQS | -.245 | -.081 | .012 |

Statistically significant correlations (P<0.05) are shown in bold.

have had to perform perimetry with a high mental load due to concentrating deeply on completing the test. On the other hand, severe sleepiness causes longer blink durations and a decreased blink rate [29–31]. Therefore, some patients whose blink rates decreased during perimetry may have had severe sleepiness. In both patients with DE and normal control subjects, VB score was worsened in some eyes with blink rates that did not decrease during perimetry (Fig 3), as well as in eyes with decreased blink rates during perimetry (Fig 3). Not only the decreased blink rate, but also the abnormal blinking caused by strong tension or severe sleepiness during perimetry may be reasons for the worsening of ocular surface conditions. Perimetrists should exercise caution and create a relaxed atmosphere in order to allow patients to maintain a high FL rate during perimetry. It is necessary that they guide patients who have a high degree of tension and severely suppress blinking during perimetry. If patients experience an increase in blinking time and decreased rate of blinking, it is advised that perimetry be temporarily postponed to allow a period of rest.

SM is a promising method for assessing tear meniscus volume, and values greater than 5 mm have been proposed as normal values [16]. In patients with DE, SM rates of 47 eyes (97.7%) were less than 5 mm; anti-glaucoma eye drop use and older age may have induced a lower tear meniscus volume. Tear meniscus volume increases with delayed blinking [32, 33], and TFBUT becomes longer in eyes with larger tear meniscus volumes [33]. Therefore, TFBUT may have been prolonged in patients with DE after perimetry. However, in patients with DE, TFBUT was prolonged after perimetry in 6 eyes (75.0%) with blinking rates that did not decrease during perimetry (Fig 2). Abnormal blinking during perimetry may be one reason for the prolonging of TFBUT. More detailed investigations are necessary to clarify the relationship between TFBUT variation and blinking pattern.

SM value was directly correlated with FN rate, and the DEQS was negatively correlated with FN rate in patients with DE. Patients with DE whose tear meniscus volumes were relatively larger, or those who exhibited less severe symptoms associated with DE, had higher FN rates. While this suggests that tear meniscus volume and severity of symptoms associated with DE may have been related to FN response in patients with DE, further studies are required to establish causation.

Patient inconsistency produces a high FN rate. This may occur if the patient's criteria changes regarding whether a stimulus is too dim to report seeing it, or there is a change in the patient's state of alertness. Additionally, there may be a lack of response from the patient due to a belief that it is too late to respond after the light goes off. High FN rates may also be produced by patients who are reliable perimetry subjects, but who have abnormal fields because visibility of near-threshold stimuli is highly variable at abnormal locations [11]. However, at present, no published studies have examined whether there is a relationship between tear meniscus volume, severity of symptoms associated with DE, and FN response. It is difficult to hypothesize the reason patients with DE whose tear meniscus volumes were large, and whose DE-associated symptoms were not severe, had higher FN rates. Further research is needed to investigate the relationship between tear metrics, blinking pattern, ocular surface conditions, and perimetry reliability using larger-scale studies. Moreover, this study did not investigate the actual field loss pattern and the clinical decision-making based on this, nor the true impact of ocular surface disease on glaucoma management. Therefore, future studies are required to investigate these problems.

## Conclusions

Performing perimetry was associated with a significant change in tear conditions and deterioration in fluorescein staining of the ocular surface in both patients with DE and normal control

subjects. Tear metrics and ocular surface conditions may impact the reliability of perimetry in patients with DE. Therefore, reliable perimetry measurements rely on appropriate evaluation of the ocular surface and tear conditions. Perimetrists should monitor blinking patterns during perimetry.

## Author Contributions

**Conceptualization:** Hideto Sagara, Hiroaki Shintake, Urara Sugiyama, Hiroki Maehara.

**Data curation:** Hideto Sagara, Tetsuju Sekiryu, Hiroaki Shintake, Urara Sugiyama, Hiroki Maehara.

**Formal analysis:** Hideto Sagara, Tetsuju Sekiryu, Hiroki Maehara.

**Funding acquisition:** Hideto Sagara.

**Investigation:** Hideto Sagara, Tetsuju Sekiryu, Kimihiro Imaizumi, Hiroaki Shintake, Urara Sugiyama, Hiroki Maehara.

**Methodology:** Hideto Sagara, Tetsuju Sekiryu, Urara Sugiyama, Hiroki Maehara.

**Project administration:** Hideto Sagara, Kimihiro Imaizumi, Hiroaki Shintake, Hiroki Maehara.

**Resources:** Hideto Sagara, Kimihiro Imaizumi.

**Software:** Hideto Sagara.

**Supervision:** Kimihiro Imaizumi, Hiroaki Shintake, Urara Sugiyama.

**Validation:** Urara Sugiyama, Hiroki Maehara.

**Visualization:** Hideto Sagara, Hiroki Maehara.

**Writing – original draft:** Hideto Sagara.

**Writing – review & editing:** Hideto Sagara, Tetsuju Sekiryu.

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
