## [Decision Letter · Decision Letter 0]

31 Jul 2019

PONE-D-19-15509

Impact of tear dynamics on the reliability of perimetry in patients with dry eye

PLOS ONE

Dear Dr. Sagara,

Thank you for submitting your manuscript to PLOS ONE. After careful consideration, we feel that it has merit but does not fully meet PLOS ONE’s publication criteria as it currently stands. Therefore, we invite you to submit a revised version of the manuscript that addresses the points raised during the review process.

We would appreciate receiving your revised manuscript by Sep 14 2019 11:59PM. To enhance the reproducibility of your results, we recommend that if applicable you deposit your laboratory protocols in protocols.io, where a protocol can be assigned its own identifier (DOI) such that it can be cited independently in the future. For instructions see: http://journals.plos.org/plosone/s/submission-guidelines#loc-laboratory-protocols

We look forward to receiving your revised manuscript.

Kind regards,

James Wolffsohn, PhD

Academic Editor

PLOS ONE

Journal Requirements:

Additional Editor Comments (if provided):

There are some significant issues to address including further data analysis for this research to be published in PlosOne so please respond to each of the reviewers comments carefully.

Reviewers' comments:

Reviewer's Responses to Questions

**Comments to the Author**

1. Is the manuscript technically sound, and do the data support the conclusions?

Reviewer #1: Partly

Reviewer #2: Partly

2. Has the statistical analysis been performed appropriately and rigorously? 

Reviewer #1: Yes

Reviewer #2: No

3. Have the authors made all data underlying the findings in their manuscript fully available?

Reviewer #1: Yes

Reviewer #2: Yes

4. Is the manuscript presented in an intelligible fashion and written in standard English?

Reviewer #1: Yes

Reviewer #2: Yes

5. Review Comments to the Author

Reviewer #1: I read this article with interest. Although, I would like to bring to your attentions some comments and thoughts I had while reviewing your manuscript.

May I begin saying that I suggest to find an alternative title as I don't agree that the research has evaluated the dynamics of the tear film. In fact, if we consider the Oxford English Dictionary, the term "dynamics" is explained as follow: "dynamics [uncountable] the science of the forces involved in movement (belongs to fluid dynamics)". For this, reason I think "tear dynamics" can be replaced with "tear characteristics" or "tear metrics" or "tear properties".

Secondly, in the "Examinations" section (line 98 and further) it is not clearly defined in which order the "tear dynamics and ocular surface condition" have been performed. As you know, some of the tear tests performed may affect the measurements because of their invasiviness (e.g. TFBUT, Schirmer test, etc.). Additionally, it is not clear how the "expert perimetrists" counted the number of blinks BEFORE perimetry (e.g. manual counting, click-counting with electronic device, click-counting using an external device such as eye tracker, etc.).

In the results section, I found there are the first 5 sentences (lines from 126 to 130) which are describing the subjects. I believe this should be moved in the participants section.

Table 2 caption (line 144 and 145) says: "Comparison of Blink Rate Parameters during Perimetry etc" but there are also included other tear metrics, not only the blink rate parameters.

Line 36: it says medications but I believe you want to say "lubricants" or "eyedrops".

Line 72: could you please define long=term follow-ups: days, weeks, months or years?

Line 85: could you please clarify what does it mean when you say "we only examined eyes in which perimetry were initally performed?" Is this an inclusion criteria?

Line 186 "ocular surface disorder", in this sentence should be "ocular surface parameters".

Lines 172, 186, 226, 236 please specify the sentence that says "some eyes" in number or percentage.

Lines 180, 181,205 please specify the standard deviations for those measurements.

Line 232 please specify "low" in number or percentage.

Lines from 242 to 245 and lines from 153 to 155: these short paragraphs are hard to follow, could you please rephrase them?

Line 258 it says medications but I believe you want to say "lubricants" or "eyedrops".

Line 225 a verb is missing where it does say "and not to decrease their...".

Line 231, strip meniscometry is not a non-invasive test as provoke tearing and measures tear reflex volume.

Line 212 "older age" should be "in elderly patients".

Line 110: this is just a suggestion. In order to improve consistency along the manuscript, I suggest to use all numbers when you mention "number of eyes" (e.g. line 110 says "eight eyes" while line 111 says "35 eyes"). Same in line 126 wherer it does say "twenty-seven eyes" and after "29 eyes").

Reviewer #2: The effect of perimetry on the ocular surface and the effect of the ocular surface on the data collected in perimetry is of interest, but the title only covers one of these aspects. What is critical is not the reliability indicators, but the actual field loss pattern and clinical decision making based on this. The authors should consider how to analyse this to understand the true impact of ocular surface disease on glaucoma management.

The abstract is hard to read without sub-headings.

Ln17 “specific effects” is not clear

Ln21 better to state “Forty-three patients with symptoms and an unstable tear film (DE according to the 2016 Japanese diagnostic criteria)…” why not use the more recent global diagnostic consensus?

Ln24 “fluorescein staining” would be clearer to the reader than “Bijsterveld score”. Why were symptoms measured before and after perimetry such as using the SANDE?

Ln28 why would TFBUT increase after perimetry? – state what this is compared to

Ln34 this is not a conclusion drawn from the data so remove – also relevant to Ln259. It can be in the discussion though as a possible remedy to the effects observed

Avoid personal pronouns such as “we” throughout

Ln62 The sample size needs to be justified

Ln100 explain how TFBUT was measured. It needs to be much clearer what were the baseline tests of ocular surface health and what were the characteristics studied to determine change with the perimetry

Ln104 “optometrists” misspelt

Ln105 were the participants unaware their blink rate was being observed

Ln108 and beyond. The study is almost certainly not powered to break the analysis down into groups as small as n=8. This should be removed throughout.

Ln115 was the data not normally distributed resulting in non-parametric tests being applied?

Ln127 it is essential to know whether the anti-glaucoma drops were preserved or not and to do an analysis of ocular surface parameters and the effect of perimetry base on this if the sample split size is sufficient

Do not repeat data which is rightly in the tables in the text, which currently makes it much harder to read

Ln221 high and low are relative terms so should be higher and lower

Ln228 explain what is meant about pattern of blinking.

More discussion is needed of the trade off between higher concentration leading to less blinking (and less likely to miss a stimulus), but then more ocular irritation leading to a loss of concentration.

Ln239-242 these sentences are unclear – the fact that metrics are associated does not mean one causes the other

6. PLOS authors have the option to publish the peer review history of their article (what does this mean?). If published, this will include your full peer review and any attached files.

Reviewer #1: No

Reviewer #2: No

---

## [Author Response · Author response to Decision Letter 0]

27 Aug 2019

August 27, 2019

Editorial board of ophthalmology

PLOS ONE

Dear Editors:

Re: “Impact of tear dynamics on the reliability of perimetry in patients with dry eye” by Tetsuju Sekiryu, Kimihiro Imaizumi, Hiroaki Shintake, Urara Sugiyama, and Hiroki Maehara (PONE-D-19-15509)

We are most grateful to the reviewers for their helpful comments on the original version of our manuscript. We have taken all these comments into account and submit the revised version of our paper.

Particularly, in accordance with the important suggestion given by reviewer #2, we rewrote the lines from 217 to 232 in the revised version of our manuscript, and quoted some references that investigated the relationships between perimetry, blink frequency, concentration, and sleepiness. We were able to identify several important relationships.

We have addressed all the comments made by reviewers #1 and #2 and have provided point-by-point responses below. We hope that our explanations and revisions are satisfactory.

We hope that the revised version of our paper is now suitable for publication in PLOS ONE, and we look forward to hearing from you at your earliest convenience.

Yours sincerely,

Hideto Sagara, MD 

We are grateful to reviewer #1 and reviewer #2 for the critical comments and suggestions that have helped us to improve our paper. As indicated in the responses that follow, we have taken all these comments and suggestions into account in the revised version of our manuscript.

Reviewer #1

Comment #1:

I read this article with interest. Although, I would like to bring to your attentions some comments and thoughts I had while reviewing your manuscript.

May I begin saying that I suggest to find an alternative title as I don't agree that the research has evaluated the dynamics of the tear film. In fact, if we consider the Oxford English Dictionary, the term "dynamics" is explained as follow: "dynamics [uncountable] the science of the forces involved in movement (belongs to fluid dynamics)". For this, reason I think "tear dynamics" can be replaced with "tear characteristics" or "tear metrics" or "tear properties".

Response

In the revised version of our manuscript, upon your advice, we have now changed the term "tear dynamics" to "tear metrics" or "tear condition" in the title, and throughout the text.

Comment #2:

Secondly, in the "Examinations" section (line 98 and further) it is not clearly defined in which order the "tear dynamics and ocular surface condition" have been performed. As you know, some of the tear tests performed may affect the measurements because of their invasiviness (e.g. TFBUT, Schirmer test, etc.). Additionally, it is not clear how the "expert perimetrists" counted the number of blinks BEFORE perimetry (e.g. manual counting, click-counting with electronic device, click-counting using an external device such as eye tracker, etc.).

Response 

Thank you for your suggestion. As some of the tear tests were invasive, the TFBUT and van Bijsterveld scores were evaluated 30 minutes before starting perimetry, and the Schirmer’s test and lacrimal irrigation were performed on other days, within 20 days of perimetry. We rewrote a sentence in the revised version of our paper in lines 107 to 108:

“Schirmer’s test without topical anesthesia and lacrimal irrigation were performed on other days, within 20 days of perimetry.”

As you have mentioned, the method of counting the number of blinks before perimetry is unclear. Therefore, we rewrote a sentence in the revised version of our paper, in lines 110 to 111 as follows:

“Expert optometrists observed the eyes of subjects and manually counted the number of blinks before perimetry, while subjects were in a relaxed atmosphere.”

Comment #3:

In the results section, I found there are the first 5 sentences (lines from 126 to 130) which are describing the subjects. I believe this should be moved in the participants section.

Response

In accordance with your suggestion, the 5 sentences have been moved to the participants section in the revised version of our paper (lines 79 to 83). 

Comment #4:

Table 2 caption (line 144 and 145) says: "Comparison of Blink Rate Parameters during Perimetry etc" but there are also included other tear metrics, not only the blink rate parameters.

Response

We have changed "Comparison of Blink Rate Parameters during Perimetry etc." to "Comparison of Blink Rate and other Parameters during Perimetry in Patients with Dry Eye and Normal Control Subjects."

Comment #5:

Line 36: it says medications but I believe you want to say "lubricants" or "eyedrops".

Response 

In accordance with your suggestion, we have now changed the term "medication" to "eye drops" throughout the manuscript.

Comment #6:

Line 72: could you please define long=term follow-ups: days, weeks, months or years?

Response

In this study, long term follow-up refers to a time period greater than 6 months. Therefore, we rewrote the sentence as follows in the revised version of our paper (lines 71 to 73): 

“Patients with DE and normal control subjects who had a follow-up of more than 6 months with at least three perimetry measurements, a best-corrected visual acuity of 20/25 or better, an intraocular pressure of ≤21 mmHg, and who were diagnosed or suspected of having glaucoma were included.” 

Comment #7:

Line 85: could you please clarify what does it mean when you say "we only examined eyes in which perimetry were initally performed?" Is this an inclusion criteria?

Response

In accordance with your suggestion, this is one of the inclusion criteria. Therefore, we moved this sentence to lines 74 to 75, in the revised version of our manuscript.

Comment #8:

Line 186 "ocular surface disorder", in this sentence should be "ocular surface parameters".

Response 

In accordance with your suggestion, we changed "ocular surface disorder" to "ocular surface parameters" or "VB score" in the revised version of our paper (lines 188, 212, and 225).

Comment #9:

Lines 172, 186, 226, 236 please specify the sentence that says "some eyes" in number or percentage.

Response

In accordance with your suggestion, we have now provided the number of eyes (and percentage) in the revised version of our paper in lines 189 and 238. We have removed sentences in lines 172 and 236.

Comment #10:

Lines 180, 181,205 please specify the standard deviations for those measurements.

Response 

Thank you for this comment. These were non-parametric data, as the Kolmogorov–Smirnov test revealed that these parameters did not follow a normal distribution. Therefore, median (IQR [min-max]) were used.

Comment #11:

Line 232 please specify "low" in number or percentage.

We rewrote lines 234 to 235, and added a sentence in the results section (lines 136 to 137) as follows:

“In patients with DE, SM rates of 47 eyes (97.7%) were less than 5 mm; anti-glaucoma eye drop use and older age may have induced a lower tear meniscus volume.” 

“In the DE group, only 1 eye (2.3%) had a normal SM rate (>5 mm), and in the control group, only one 1 eye (2.3%) also had a normal SM rate.” 

Comment #12:

Lines from 242 to 245 and lines from 153 to 155: these short paragraphs are hard to follow, could you please rephrase them?

Response

In accordance with your advice, we have rephrased the sentences in the revised version of our paper as follows: 

(lines 247 to 250)

“This may occur if the patient's criteria changes regarding whether a stimulus is too dim to report seeing it, or there is a change in the patient's state of alertness. Additionally, there may be a lack of response from the patient due to a belief that it is too late to respond after the light goes off.”

(lines 160 to 162)

“Blink rates during perimetry were significantly lower in Group 1 compared to Group 2, and in Group 3 compared to Group 4: Group 1, 4.6 (1.7-7.0 [0-21]) vs. Group 2, 13.4 (11.4-16.0 [2-39]) /min and Group 3, 2.6 (0.9-5.4 [0-12]) vs. Group 4, 20.1 (11.0-30.5 [4-36]) /min (P=0.007 and P<0.001, respectively, Fig 1).”

Comment #13

Line 258 it says medications but I believe you want to say "lubricants" or "eyedrops".

Response

In accordance with your suggestion, we revised "medication" to "eye drops."

Comment #14:

Line 225 a verb is missing where it does say "and not to decrease their...".

Response

In accordance with your advice, we rewrote the sentence in lines 228 to 229 as follows:

“Perimetrists should exercise caution and create a relaxed atmosphere in order to allow patients to maintain a high FL rate during perimetry.” 

Comment #15:

Line 231, strip meniscometry is not a non-invasive test as provoke tearing and measures tear reflex volume.

Response

Thank you for this comment. We have re-written the sentence in lines 233 to 234 as follows:

“SM is a promising method for assessing tear meniscus volume, and values greater than 5 mm have been proposed as normal values [16].” 

Comment #16:

Line 212 "older age" should be "in elderly patients".

Response

In accordance with your suggestion, we rewrote the sentence in lines 206 to 207 as follows:

“Several studies have indicated that the use of anti-glaucoma eye drops increased the risk of DE [23, 24], and that the rate of DE in elderly patients was higher [25, 26].” 

Comment #17:

Line 110: this is just a suggestion. In order to improve consistency along the manuscript, I suggest to use all numbers when you mention "number of eyes" (e.g. line 110 says "eight eyes" while line 111 says "35 eyes"). Same in line 126 wherer it does say "twenty-seven eyes" and after "29 eyes").

Response

In accordance with your advice, we have now made this modification when referring to the number of eyes: 

(lines 79 to 80)

“A total of 27 eyes (62.8%) and 29 eyes (67.4%) in patients with DE, and normal control subjects, respectively, were treated with anti-glaucoma eye drops.” 

(lines 118 to 119)

“Group 2 comprised 8 eyes in 8 patients with DE with blink rates that increased or remained unchanged during perimetry.”

(lines 121 to 122)

“Group 4 comprised 8 eyes in 8 normal control subjects with blink rates that increased or remained unchanged during perimetry.”

(lines 140 to 141)

“Blink rates of 35 eyes (81.4%) decreased, and the rates of 8 eyes (18.6%) increased or remained unchanged during perimetry in both patients with DE and normal control subjects.” 

We are grateful to reviewer #2 for the critical comments and important advice that have helped us to improve our paper. As indicated in the responses that follow, we have taken all these comments and suggestions into account in the revised version of our paper.

Reviewer #2: 

Comment #1:

The effect of perimetry on the ocular surface and the effect of the ocular surface on the data collected in perimetry is of interest, but the title only covers one of these aspects. What is critical is not the reliability indicators, but the actual field loss pattern and clinical decision making based on this. The authors should consider how to analyse this to understand the true impact of ocular surface disease on glaucoma management.

Response

Thank you for this suggestion. We have now changed our manuscript title to "Impact of tear metrics on the reliability of perimetry in patients with dry eye". We did not investigate the actual field loss pattern, clinical decision making based on this, or the true impact of ocular surface disease on glaucoma management. Therefore, we will investigate these problems in a future study, and we have added a sentence in the revised version of our paper (line 256) to reflect this.

“Moreover, this study did not investigate the actual field loss pattern and the clinical decision-making based on this, nor the true impact of ocular surface disease on glaucoma management. Therefore, future studies are required to investigate these problems.”

Comment #2:

The abstract is hard to read without sub-headings.

Response 

In accordance with your advice, we rewrote our abstract with sub-headings to facilitate easier understanding.

Comment #3:

Ln17 “specific effects” is not clear

Response

In line with your suggestion, we rewrote “the specific effects” as “the effects” in lines 17, 45, and 51.

Comment #4:

Ln21 better to state “Forty-three patients with symptoms and an unstable tear film (DE according to the 2016 Japanese diagnostic criteria)…” why not use the more recent global diagnostic consensus?

Response

In 2016, the Asia Dry Eye Society and the Dry Eye Society of Japan implemented new diagnostic criteria for dry eye disease, and at the present time, almost all investigators in Japan use these criteria to diagnose dry eye. This definition clearly covers the short TFBUT-type dry eye, the major type of dry eye, and emphasizes the importance of measuring tear film stability by TFBUT.

Comment #5:

Ln24 “fluorescein staining” would be clearer to the reader than “Bijsterveld score”. Why were symptoms measured before and after perimetry such as using the SANDE?

Response

In accordance with your advice, we have now changed the “Bijsterveld score” to “fluorescein staining” in lines 24, 26 and 30. SANDE is useful, as well as the Ocular Surface Disease Index (OSDI) and DEQS, to diagnose dry eye. However, the DEQS is more popular than SANDE in Japan. The DEQS can assess various aspects of Quality of Life including its mental aspect. The Dry Eye Society of Japan recommends using the DEQS to diagnose dry eye. Therefore, we used the DEQS in this study.

Comment #6:

Ln28 why would TFBUT increase after perimetry? – state what this is compared to

Response

We rewrote the sentence as follows (lines 29 to 30):

“TFBUT after perimetry was significantly higher than before perimetry in patients with DE (P<0.001).”

Comment #7:

Ln34 this is not a conclusion drawn from the data so remove – also relevant to Ln259. It can be in the discussion though as a possible remedy to the effects observed

Avoid personal pronouns such as “we” throughout

Response

In accordance with your suggestion, we removed these sentences from the conclusions in lines 35 and 266. We removed “we” and rewrote the sentences in lines 48, 56, 74, 93, 94 and 103 as follows:

(line 48)

“As a result of this concentrated viewing (i.e., “staring”), tear condition may change drastically during perimetry.”

(line 56)

“These indices were compared in two groups: patients with DE, and normal control subjects.”

(line 74)

“Only eyes in which perimetry was initially performed were examined.”

(line 93)

“In this study, DE was diagnosed according to the new diagnostic criteria.”

(line 94)

“Dry Eye-related Quality of Life Score (DEQS) questionnaires were used to evaluate the symptoms associated with DE [15].”

(line 103)

“Tear metrics and ocular surface condition were evaluated using five parameters:”

Comment #8:

Ln62 The sample size needs to be justified

Response

In this study, all patients attending the glaucoma specialty outpatient clinic of Fukushima Medical University or the Marui Eye Clinic from May to November 2014 were considered, and we justified the sample size of dry eye patients and normal control subjects. However, the sample size was relatively small, so in the future, we will investigate this using larger-scale studies. We have changed a sentence in lines 66-68 as follows:

“All patients attending the glaucoma specialty outpatient clinic of Fukushima Medical University or the Marui Eye Clinic from May to November 2014 were considered.”

Comment #9:

Ln100 explain how TFBUT was measured. It needs to be much clearer what were the baseline tests of ocular surface health and what were the characteristics studied to determine change with the perimetry

Response

In accordance with your suggestion, we have described how TFBUT was measured in lines 112 to 114 as follows:

“For measurement of TFBUT, the interval between the last complete blink and the first appearance of a dry spot or disruption in the tear film was recorded after the instillation of fluorescein. TFBUT was evaluated three times, and the mean value was recorded.” 

Comment #10:

Ln104 “optometrists” misspelt

Response

In accordance with your suggestion, we have changed “perimetrists” to“optometrists” in line 110.

Comment #10:

Ln105 were the participants unaware their blink rate was being observed

Response 

We have added a sentence in lines 111 to 112 as follows:

“The participants were unaware that their blink rate was being observed.”

Comment #11:

Ln108 and beyond. The study is almost certainly not powered to break the analysis down into groups as small as n=8. This should be removed throughout.

Response

Thank you for your comment. We acknowledge that the number of dry eye patients is small. However, we researched more than 250 patients and found only eight patients with dry eye and blink rates that increased or remained unchanged during perimetry. In addition, we found a significant difference in fixation loss late. Therefore, while acknowledging the low patient number, we believe the results are still of great interest, and can serve to inform future research in this area. 

Comment #12:

Ln115 was the data not normally distributed resulting in non-parametric tests being applied?

Response

Measurements, other than age and test time, were found to be non-parametric according to the Kolmogorov–Smirnov test which revealed that these parameters did not follow a normal distribution. Therefore, the median (IQR [min-max]) was reported for these parameters, and the Wilcoxon signed-rank test and the Mann-Whitney U test were used for statistical analysis.

Comment #13:

Ln127 it is essential to know whether the anti-glaucoma drops were preserved or not and to do an analysis of ocular surface parameters and the effect of perimetry base on this if the sample split size is sufficient

Response 

In accordance with your suggestion, we rephrased lines 106-107 and lines l08-110 as follows:

(lines 106-107)

“The use of all eye drops, including anti-glaucoma eye drops, was stopped 4 hours before all examinations to avoid eye drop effects.”

(lines l08-110) 

“SM value, blink rate in 1 min, TFBUT, and VB scores were evaluated 30 mins before starting perimetry. TFBUT and VB scores were re-evaluated immediately after the first perimetry.” 

Comment #14:

Do not repeat data which is rightly in the tables in the text, which currently makes it much harder to read

Response 

We removed the repeated data from lines 176 and 206.

Comment #15:

Ln221 high and low are relative terms so should be higher and lower

Response

We corrected the sentence in lines 219 to 220 as follows:

“The blinking rate is lower when mental load is higher. Conversely, the blinking rate is higher when mental load is lower [22].”

Comment #16:

Ln228 explain what is meant about pattern of blinking.

Response

This sentence is unclear, therefore, we rewrote the sentence in lines 217 to 219 as follows:

“A wide range of blink frequencies occur during routine perimetry. Although some patients hardly ever blink during a perimetry, others show frequent blinking that often appears to be related to stimulus presentation times [28].”

Comment #17:

More discussion is needed of the trade off between higher concentration leading to less blinking (and less likely to miss a stimulus), but then more ocular irritation leading to a loss of concentration.

Response

We rewrote lines 217 to 232, and quoted some references which have investigated the relationships between perimetry, blink frequency, concentration and sleepiness as follows: 

“A wide range of blink frequencies occur during routine perimetry. Although some patients hardly ever blink during perimetry, others show frequent blinking that often appears to be related to stimulus presentation times [28]. The blinking rate is lower when mental load is higher. Conversely, the blinking rate is higher when mental load is lower [22]. The patients whose blinking rates did not decrease during perimetry may have been able to perform perimetry while relaxed, whereas patients whose blinking rates decreased during perimetry may have had to perform perimetry with a high mental load due to concentrating deeply on completing the test. On the other hand, severe sleepiness causes longer blink durations and a decreased blink rate [29-31]. Therefore, some patients whose blink rates decreased during perimetry may have had severe sleepiness. In both patients with DE and normal control subjects, VB score was worsened in some eyes with blink rates that did not decrease during perimetry, as well as in eyes with decreased blink rates during perimetry (Fig 3). Not only the decreased blink rate, but also the abnormal blinking caused by strong tension or severe sleepiness during perimetry may be reasons for the worsening of ocular surface conditions. Perimetrists should exercise caution and create a relaxed atmosphere in order to allow patients to maintain a high FL rate during perimetry. It is necessary that they guide patients who have a high degree of tension and severely suppress blinking during perimetry. If patients experience an increase in blinking time and decreased rate of blinking, it is advised that perimetry be temporarily postponed to allow a period of rest.”

Comment #18:

Ln239-242 these sentences are unclear – the fact that metrics are associated does not mean one causes the other

Response

We have modified this section as follows:

(lines 242 to 246)

“SM value was directly correlated with FN rate, and the DEQS was negatively correlated with FN rate in patients with DE. Patients with DE whose tear meniscus volumes were relatively larger, or those who exhibited less severe symptoms associated with DE, had higher FN rates. While this suggests that tear meniscus volume and severity of symptoms associated with DE may have been related to FN response in patients with DE, further studies are required to establish causation.”

---

## [Editor Report · Decision Letter 1]

30 Aug 2019

Impact of tear metrics on the reliability of perimetry in patients with dry eye

PONE-D-19-15509R1

Dear Dr. Sagara,

We are pleased to inform you that your manuscript has been judged scientifically suitable for publication and will be formally accepted for publication once it complies with all outstanding technical requirements.

With kind regards,

James Wolffsohn, PhD

Academic Editor

PLOS ONE

Additional Editor Comments:

Thank you for addressing the reviewers comments

---

## [Editor Report · Acceptance letter]

10 Sep 2019

PONE-D-19-15509R1 

Impact of tear metrics on the reliability of perimetry in patients with dry eye 

Dear Dr. Sagara:

I am pleased to inform you that your manuscript has been deemed suitable for publication in PLOS ONE. Congratulations! Your manuscript is now with our production department. 

With kind regards,

on behalf of

Professor James Wolffsohn 

Academic Editor

PLOS ONE